# A Biomimetic Method to Replicate the Natural Fluid Movements of Swimming Snakes to Design Aquatic Robots

**DOI:** 10.3390/biomimetics7040223

**Published:** 2022-12-03

**Authors:** Elie Gautreau, Xavier Bonnet, Juan Sandoval, Guillaume Fosseries, Anthony Herrel, Marc Arsicault, Saïd Zeghloul, Med Amine Laribi

**Affiliations:** 1Department of GMSC, Pprime Institute, University of Poitiers, CNRS, ISAE-ENSMA, UPR 3346 Poitiers, France; 2CEBC Center of Biological Studies of Chizé, CNRS & University of La Rochelle, Villiers-en-Bois, UMR 7372 Deux-Sèvres, France; 3MNHN National Museum of Natural History, CNRS, UMR 7179 Paris, France

**Keywords:** snake robot, biomimicry, compliant mechanism, design, motion analysis

## Abstract

Replicating animal movements with robots provides powerful research tools because key parameters can be manipulated at will. Facing the lack of standard methods and the high complexity of biological systems, an incremental bioinspired approach is required. We followed this method to design a snake robot capable of reproducing the natural swimming gait of snakes, i.e., the lateral undulations of the whole body. Our goal was to shift away from the classical broken line design of poly-articulated snake robots to mimic the far more complex fluid movements of snakes. First, we examined the musculoskeletal systems of different snake species to extract key information, such as the flexibility or stiffness of the body. Second, we gathered the swimming kinematics of living snakes. Third, we developed a toolbox to implement the data that are relevant to technical solutions. We eventually built a prototype of an artificial body (not yet fitted with motors) that successfully reproduced the natural fluid lateral undulations of snakes when they swim. This basis is an essential step for designing realistic autonomous snake robots.

## 1. Introduction

Underwater robotics is an active research field [1,2,3]. Reproducing animal movements offers a means for exploring the respective roles of key elements of the skeleton–muscular system, gait, and kinematics because selected parameters can be manipulated, and thus, disentangled [4]. Conversely, unraveling the locomotor complexity and diversity of animals provides inspiration to roboticists [5]. Thus, bio-inspired robots are useful tools for various, albeit complementary, types of research. For instance, the aquatic snake robot Amphibot II was developed using neuroscience: the control of locomotion was performed using a central pattern generator (CPG) [1,2,3]. Paleontology benefited from the development of a fossil robot [4]. The field of practical applications using bio-inspired underwater robots is rapidly growing, notably for technical, environmental, and contamination monitoring [5,6].

Among autonomous underwater vehicles (UVs), fish robots occupy a central place [7]. Usually, fish robots use planar oscillatory locomotion, such as the carangiform and subcarangiform swimming gaits observed in living fish, where the tail produces a propelling thrust [6,7,8,9]. The tail beats are fluid-driven [10], cable-driven [11], or magnetically actuated [12]. Actuation is coupled with flexible material for compliant fluid motion to mimic living fish. The compliant fins are generally made of silicone rubber [13], while the body is made of several rigid segments [14,15]. By contrast, in anguilliform swimming locomotion (e.g., that observed in eels), thrust is produced by the whole body; it is a poorly studied gait in robotics. However, anguilliform swimming is particularly energy-efficient, requiring four to six times less energy compared with the other fish swimming modes [8]. Snakes also use the lateral undulation of the whole body to swim (anguilliform gait), and thus, they represent a source of inspiration. Underwater snake robots are mostly bio-inspired by snakes [16], eels [17], lampreys [2], and salamanders [18]. Anguilliform locomotion that relies on the continuous deformation of the whole body makes the design of such robots far more complex than classical and simpler fish robots. Many bio-inspired snake robots were recently developed following the seminal input of Shiego Hirose [19]. Two categories of snake robots are identified: (1) snake manipulator robots, for instance, those used for mini-invasive surgery, frequently involve cable-driven actuation where servomotors are concentrated at the robot base [20], and (2) underwater snake robots, which were developed using a modular design of identical segments with pivot or universal rigid joints that actively or passively actuate as passive torsion joints [1]. However, the undulations result from the actuation of a limited number of rigid segments (usually less than 20); the undulatory kinematics follow a broken line [21]. This sharply contrasts with the natural fluid movement of living snakes that are made of hundreds of articulated vertebrae (typically 200 to 300). The lack of fluidity of snake robots hampers the mimicking of real snake movements, and thus, this poses difficulties in accurately studying the importance of body wavelength, shape, frequency, and amplitude, as well as the propagation of undulations from the head to the tail. Therefore, an alternative approach is needed to design more realistic snake robots that are capable of generating fluid undulations. One option is to ground this approach on the anatomical observations of snakes and the swimming monitoring of swimming snakes. Collaborations with biologists are then essential to create an effective design of underwater snake robots.

So far, snake-like robot actuation was mostly achieved with DC motors [22] and servomotors [16]. Connecting the modules serially allows the snake-like robot to swim in a plane [1]. The combination of orthogonal joints enables swimming in a volume [16]. A combination of actuators for pitch, yaw, and active torsion at each joint was proposed by Boyer et al. to increase the fluidity [17]. Various actuation techniques, such as magnetic actuation [2], were investigated. The design of such robots tends to include artificial muscles [23] to obtain compliance and improve the motion fluidity. Additionally, pneumatic actuation is applied to achieve planar locomotion [24]. No cable-driven actuation has been used to design autonomous snake robots (to the best knowledge of the authors); however, this technique provides particularly fluid movements in snake manipulator robots.

Another reason to not use a few repeated identical rigid segments [16,22] to conceive a snake robot is that the morphology of snakes varies from the snout to the tip of the tail, with larger sections occurring mid-body, and the snake’s body shape being able to change during swimming [25]. More generally, the natural snake flexibility of the complex musculoskeletal system is poorly introduced in the design of such robots [26]. In this study, we further developed a biomimetic approach based on compliant bio-inspired modules to design realistic swimming snake robots [27]. This method involved shifting away from rigid modules to develop a fully flexible robotic structure with passive behavior close to what can be observed in real snakes. The objective was to build a scientific tool designed to study the undulatory swimming of snakes. We limited our study to unidirectional swimming (i.e., a snake swimming in a straight line); thus, the great maneuverability of living snakes was not considered in this study. We also emphasize that we also limited this study to the conception and testing of an artificial snake body structure. We did not fit this structure with motors plus movement controllers; thus, we did not attempt to construct a fully autonomous waterproof aquatic snake robot. Nevertheless, a body structure capable of reproducing the natural movements of snakes represents a crucial prerequisite to designing a realistic snake robot. Realizing the natural curvature and fluid motions was made possible by the emergence of soft robotics based on the use of cable-driven continuum manipulators and a series of compliant joints [28]. The combination of flexible materials and rigid links allows for movements to be obtained that correctly mimic those that are observed in biological systems [15]. Therefore, we collected fundamental information on the anatomy and the swimming kinematics of snakes to conceive an artificial snake body structure made of serial but not identical compliant joints in order to mimic the complex morphologies of snakes. Importantly, the rigidity of this structure was designed to reproduce the variations observed along a snake’s body. We then compared the results of this body structure submitted to the undulations generated by a motor attached to the “head” with those obtained in living snakes.

## 2. BIM: Bio-Inspired Method

### 2.1. Introduction to BIM

A generic method was proposed to mechanically synthesize a snake’s musculoskeletal system in order to mimic anguilliform swimming locomotion. It consisted of studying the biological system to identify the main links between mechanics and kinematics. Next, a three-step method was proposed (see Figure 1) to mimic the musculoskeletal system of snakes and to design a robot that was capable of reproducing the body kinematics of swimming snakes. It resulted in a strongly bio-inspired design that was derived from the observation of a natural system to achieve anguilliform locomotion.

**Step 1**: Observations of snake anatomy provided the initial input data. Vertebra morphology and variation along the body axis and their assemblages were first considered. The rigid body replacement method was applied to design the shape of a snake robot skeleton constrained by specific motions imposed by the vertebral anatomy. The geometric shape obtained needed to be sized.

**Step 2**: The geometric shape was sized according to snake morphology and imposed by the limits of the technical solution envisaged. These constraints were implemented as input data regarding the number of vertebrae of biological snakes ($N_{vb})$ and the number of vertebrae of robotic snakes ($N_{vr}$), combined with snake swimming kinematics monitored with specifically developed software [29] (Software for the Analysis of Anguilliform Swimming (SAAS)). A ratio $R_{v}$ between $N_{vr}$ and $N_{vb}$ was defined. The diameter of the snake robot was scaled to the diameter of the biological snake $D_{s}$. It enabled the sizing of the diameters of consecutive robot disks. The number of vertebrae, the length of a biological snake vertebra $L_{1vb}$, and the angles between the vertebrae $\alpha_{b}$ were combined to adapt the length of the snake robot vertebrae and the number of vertebrae, and to size the mechanical stop of the geometric shape $\alpha_{r}$, respectively. The result was a mechanical system that was directly sized from the biological data. 

**Step 3**: The elastic behavior of the musculoskeletal assemblages of a snake body needed to be added to the mechanical system (E, Young’s modulus; I, quadratic moment of inertia). The musculoskeletal bending moment $M_{b}$ was extracted from biological snakes and transferred to the mechanical system through a compliant universal joint. The proposed compliant universal joint design consisted of beams that were assembled in parallel for each plane. Adjusting the diameter of each beam $d_{max}$ resulted in a stiffness modification. Thus, the whole snake robot’s stiffness was computed from the biology, and the beam diameters were computed from the combination of the mechanical stop angle $\theta_{tmax}$ and the arbitrarily defined beam length $L_{pr}$. Similarly, the torsion beam moment $M_{t}$ was computed from the measured deflection angle φ. This resulted in a mechanical system with non-random elastic behavior.

Note that, theoretically, the proposed method enabled the mimicking of the musculoskeletal system of the snake by taking inspiration from any snake species. In this study, we used snake species ranging from the terrestrial/burrower python (*Python regius*) and fast terrestrial racer (*Hierophis viridiflavus*) to semi-aquatic species (*Helicops angulatus* and *Natrix helvetica*). Importantly, all of these species of snake can swim using lateral undulations. The snake robot was inspired by multiple snake species and individuals. It covered a wide range of snake behaviors and musculoskeletal systems, allowing it to be more general.

### 2.2. Step 1: Geometric Shape—The Rigid Body Replacement Method

The beam design was directly inspired by real snakes using the RBR (rigid body replacement) method [30] (see Figure 2). The RBR method was the basis that was used to introduce a flexible module (FM) that synthesized the musculoskeletal body of the snake [27]. The equivalency of the FM was proposed to size the damping, stiffness, section area, and quadratic moment of inertia that will be used for modeling and simulation in future work [22]. Starting from a snake anatomical study with the backbone, vertebrae, and rib sizes (which scale to the vertebrae and the snake diameter along the body in each species), we proposed an innovative technological shape based on a design of a universal compliant joint (see Figure 3). 

The X-ray of a python (Figure 2a) shows a backbone made of vertebrae that are relatively similar in shape but that progressively vary in size along the body axis. For simplicity, the motion between the vertebrae was modeled as a universal joint (Figure 2b) with a mechanical stop due to the high complexity of the biological backbone (Figure 2b). In practice, the real motion can be assimilated to the association of a ball-and-socket joint with a mechanical stop (bending and small torsion) and a sliding joint (small elongation), as the backbone can expand. The proposed kinematic scheme was composed of a combination of compliant universal joints associated with each other and 3D printed in a single monolithic part (Figure 2d). Mechanical stops that represent simplified pre- and post-zygapophyses (i.e., the articular processes that limit movements between adjacent vertebrae, usually two anterior and two posterior in vertebrates) were added to mimic the real vertebrae movements [31]. The compliant joint, i.e., the base element of the snake robot body, was easy to size in both lateral bending and in dorsal/ventral bending and torsion, but the latter was difficult to characterize (material density and the quadratic moment of inertia) for modeling and simulation. An equivalent model was proposed to cope with this complexity (Figure 2e). A theoretical equivalent elastic rod made of a single material and with a variable diameter was proposed for each orthogonal direction (Figure 2f). The latter was not indented to represent snake behavior, but to theoretically synthesize the 3D-printed backbone. 

### 2.3. Step 2: Sizing the Geometric Shape

As each vertebra of a backbone is unique, the angles (lateral side, dorsal, and lateral) of the mechanical stop vertebrae decreased from the neck to the body and increased from the body to the tail (see Figure 4, Figure 5 and Figure 6). The neck and tail are not only used for swimming, but also for various other movements (e.g., coiling) and maneuverability (orientation changes) during day-to-day displacements (e.g., foraging). We measured the maximum angles of the neck, middle body, and tail of three snakes: two Python regius (terrestrial) and one Helicops angulatus (semi-aquatic snake) using 2D X-rays (see Figure 4). The angles and lengths of the vertebrae (see Figure 7) were extracted using ImageJ software. In order to obtain the largest swept area, the latter was computed with the measured angles and using a constant curvature assumption [32]. The swept area was computed for five vertebrae.

To transfer information from biology to robotics, a minimum of equations was used. A vertebra ratio $R_{v}$ (Equation (1)) was applied to size the robot’s vertebra length $L_{vr_{\;}}$ (Equation (2)) according to the snake part and the total number of vertebrae $N_{vr_{\;}}$ that composed the robot (Equation (3)). Five biological vertebrae were chosen to design a bio-inspired joint (refer to Section 2.5 for the chosen number).
(1)$$R_{v}=\frac{N_{vb}}{N_{vr}}$$

$N_{vr}$ refers to the equivalent number of vertebrae for the robot; $N_{vb}$ refers to the number of vertebrae of the biological snake.
(2)$$L_{1vr_{i}}=L_{1vb_{i}}\cdot \;R_{v}\;,\;i\in \left[Neck,\;Middle\;body,\;Tail\right]$$

$L_{1vr_{\;}}$ refers to the length of one vertebra of the robot; $L_{1vb_{\;}}$ refers to the length of one vertebra of the biological snake.
(3)$$N_{vr_{i}}=\frac{N_{vb_{i}}}{R_{v}},\;i\in \left[Neck,\;Middle\;body,\;Tail\right]$$

Mechanical stops of the CU joint were sized according to the number of synthetized biological vertebrae. An equivalent angle $\alpha_{r}$ (Equation (4)) defined the amplitude of the mechanical stop of the snake skeleton.
(4)$$\alpha_{r_{i}}=\alpha_{b_{i}}\cdot \;R_{v}\;,\;i\in \left[Neck,\;Middle\;body,\;Tail\right]$$

**Remark** **1.***Bio-inspired vertebrae can be scaled from a large snake size (an adult boa or python that can exceed 50 kg) to very small species (neonates of water snakes, such as Natrix maura, that weigh less than 2 g) through the vertebrae sizes and the disk diameters. Note that an offset of disk diameter is required for the integration of actuators (a diameter of 61* mm). *In the present case, an offset of 15 mm in diameter was applied (see Figure 8). The body diameter of the robot was greater than those of whip snakes (Hierophis viridiflavus) and semi-aquatic snakes (Natrix helvetica), but it remained in the same proportion compared with terrestrial/burrower snakes, such as Python regius*.

### 2.4. Step 3: Sizing the CU Joint

Stiffness is measured on dead snakes and transferred into a compliant joint to reproduce the mechanical behavior obtained from the association of ligaments, tendons, muscles, and vertebrae that characterize the main locomotor structure of a snake [33]. The three major axial muscles (iliocostalis, semispinalis spinalis, and longissimus) are linked to the vertebrae, but also to one another [34,35]. In our tests, the skin was not taken into account when measuring the stiffness.

Five distinctive parts were selected on dead snakes. Dead snakes were collected opportunistically, for example killed by domestic cats or hit by vehicles (permit issued to XB, DBEC 004/2022). Only recently killed snakes in very good condition (i.e., skin, locomotor muscles and vertebrae of selected segments were intact) were retained. Namely, the neck (1/5 of the body), 2/5 of the body, mid-body, 3/5 of the body, and before the tail—to study the lateral side deflection under a load. Each piece was composed of 35 vertebrae (see Figure 9). The 15 first and last vertebrae were clamped. The five middle vertebrae were manually bent. The force applied was measured using a dynamometer (Lutron Force Gauge FG-20Kg-232). The beam deflection was measured with a protractor (see Figure 9). This deflection was associated with a clamped and free beam. The method was applied to five parts (1/5, 2/5, 3/5, 4/5, and 5/5 of the body) of two Hierophis viridiflavus and one Zamenis longissimus. The load and the deflection angle were interpolated (see Figure 10a,b). There was a variation in the effort and angle measurements. We took points included in the measurement intervals (see Figure 11a,b).

The applied force and the beam length allowed for the instantaneous bending moment (torque) of a large deflection non-linear elastic beam [35] (Equation (5)) to be computed and the maximum moment was determined at the clamped side (s=0).
(5)$$M_{s}=-F\cdot S\cdot \cos\varphi$$

φ defines the beam deflection at any position s of the beam, F refers to the maximum force applied on the snake body, l refers to the total beam length, and s refers to the beam position. 

The resulting equivalent bending moment $M_{b}$ (see Figure 9) was equal to the equivalent torsion moment $M_{t}{}_{max}$, as elastic beams were linked in parallel (see Figure 12b). Thus, half of the bending moment $\frac{M_{b}}{2}$ was applied on each torsion beam, i.e., $B_{1}$ and $B_{2}$ (see Figure 12a). 

The lengths of the beam and the mechanical stop, which was measured using X-rays, were imposed. Thus, the torsion angle and torsion moment defined the beam diameter $d_{max}$ (Equation (6)).
(6)$$d_{max_{i,j}}=\sqrt[{4}]{\frac{32M_{t_{max_{i,j}}}}{\pi .G.\theta_{t_{max_{i,j}}}}}$$
where $j\in \left[Lateral\;side\;1,\;lateral\;side\;2,\;Dorsal,\;Ventral\right]$ and $i\in \left[Neck,\;Middle\;body,\;Tail\right]$. $\theta_{t}$ is the unit angle of torsion (Equation (7)) and $M_{t_{max_{i,j}}}$ is the torsion moment equivalent to the bending moment extracted from the snake body at the neck, the middle of the body, and the tail.
(7)$$\theta_{t_{max_{i,j}}}=\frac{\alpha_{t_{max_{i,j}}}}{L_{pr}}$$

$\alpha_{t_{max_{\;}}}=\alpha_{r}$ is the maximal torsion angle (see Figure 12a) extracted from the X-ray measurements. This angle is given by the mechanical stop size defined between the pre- and post-zygapophyses (see Figure 12a). Mechanically constraining a movement limits the energy expended in the system due to a passive constraint and simplifies the control.

The torque applied to size the beam radius was computed for the neck, middle body, and tail in the horizontal plane (lateral sides). All radii of the beam, composed of the compliant vertebrae, were computed through a second-degree polynomial interpolation of the three measured points (head, middle body, and tail) (see Figure 13a,b).

As a result, the selected number of 6.5 vertebrae snake portions was synthetized into one compliant joint (see Figure 2). This ratio was kept along the whole body. Thus, for a biological snake of 300 vertebrae, a robotic snake will be composed of 45 artificial vertebrae. 

**Remark** **2.***The damping, stiffness, quadratic moment of inertia, diameter, Young’s modulus, and material density must be known in order to perform beam modeling (see remark 3). The pseudo-rigid body model was completed by adding the beam characteristics. As described in Figure 8, the snake body deflection was empirically evaluated through a straightforward methodology. The measured bending moment was amplified (factor of 10) to a torsion beam of printable size. In fact, with the bending moment (measured on biological snakes), the torsion beam diameter was close to a millimeter, which made it impossible to be manufactured. The non-linear beam deflection can be modeled through various beam models, such as a Timoshenko beam and the pseudo-rigid body model* [36]*. For instance, the Euler–Bernoulli model of a large deflection non-linear elastic material cantilever beam* [37,38] *can be used (see Equation (8))*. 


(8)
$$EI\varphi^{\prime \prime }+Fsin\left(\varphi \right)=0$$


I*refers to the inertia moment;* 
α *defines the force direction of the experiment*.

**Remark** **3.***Modeling the deformation of a flexible beam under external forces can be achieved through various methods. A geometrical method well known as piecewise constant curvature (PCC) is one of the most commonly used modeling methods* [32,39]*. Variable curvature modeling methods consider Young’s modulus and the quadratic moment of inertia. Dynamic methods, such as the pseudo-rigid body model (PRB), have been widely studied* [36] *to model a continuum robotic arm. The damping and stiffness of an elastic rod are taken into account for beam deflection under external forces. Cosserat rod theory is increasingly used to model flexible beam deflection under internal and external stress* [40,41,42]*. A flexible beam can be modeled as a locomotor* [22] *or a manipulator for minimally invasive surgery* [43,44]*. Internal stress is induced via cable tension in the case of cable-driven continuum robots* [45]*, pneumatic actuation* [46]*, or magnetic actuation (magnetic continuum robots)* [47,48,49,50].

**Remark** **4.***Bio-inspired vertebrae can be scaled from large snake sizes (such as boas and pythons) to smaller species (such as juvenile Natrix maura) according to the material. Flexible materials, such as PA11, PA12, and TPU, are mostly used for large snake robots (>1000 mm in length)*.

### 2.5. Bio-Inspired Snake Robot Skeletons

As a result, the successive steps of the approach lead to the development of a snake robot skeleton without actuation (see Figure 14b). The slender robot skeleton was considered an inert body that accurately reproduced the elasticity of a snake body represented by a dead skinned snake. As described in the previous sub-section, the snake robot skeleton was made up of 45 vertebrae. All of the vertebrae were attached and can be printed in one single piece. However, due to the printer dimensions, the snake robot skeleton was split into four printable parts (see Figure 14d). The snake robot was printed in Nylon (PA12) with MJF (Multi Jet Fusion) from the HP^®^ process. Each vertebra was different from the others (torsional beam diameter, mechanical stop, and vertebrae length). 

Indeed, the length of the snake robot skeleton (SnaBiBot) was in the wide range of the lengths of snakes studied here, as shown in Figure 15. This expressed the capacity of the robot to be scaled to different lengths, unlike rigid-link snake robots. 

Note that each vertebra is composed of a disk and eight holes (see Figure 14b). This had the dual function of giving a shape to the snake robot according to biological snake morphology (see Figure 8) and housing actuation cables. For the next step, the internal actuation was achieved using cables placed through the holes, allowing the flexible robot to bend, and resulting in a cable-driven continuum locomotor robot. This actuation mode is routinely used in continuum manipulators for surgery [44] and inspections [50]. However, to our knowledge, there is no cable-driven snake robot that exists. The antagonistic braided steel cables were wound in opposite directions and on a single pulley [27]. The pulley was mounted on a servomotor (Dynamixel^®^, 2XC-430-W250-T). The cables were routed through the disks to bend the compliant vertebrae (see Figure 14c). An additional description of the system is given in [27]. 

## 3. Motion Analysis and Design Validation

In this section, the snake robot’s body is compared with those of living snakes. The goal was to produce a similar behavior, including the number of waves, wavelengths, amplitude, and frequency for identical inputs. Head transverse velocity and head amplitude were the inputs used from the swimming snake motion [29,51] for the following aims: (1) to evaluate how well the compliant snake body mimicked a swimming snake, and (2) to validate the design process and mechanical design by comparing the undulation of the snake robot and the inert body of the biological snake.

### 3.1. Materials

A swimming test bench was specifically developed to analyze inert snake motions (see Figure 16). A linear axis was actuated using a stepper motor and was controlled using Arduino, allowing for the amplitudes and velocities of the snake heads to be repeated. The snake’s head was attached to the linear axis through a passive pivot link (see Figure 16). This joint enabled the body ripples to propagate the inflection point all along the snake robot’s body without losing too much energy. The snakes were positioned on a Plexiglas table. The table was sloped to assist with undulations. Moreover, glycerin was used to reduce the friction between the body and the table.

Linear axis inputs were determined through the observations made on snake undulations while swimming (see Figure 17a,b). Anguilliform snake swimming was analyzed and described using motion capture (MoCap) and video processing analysis software [29]. In the present study, the snake motions were analyzed with MoCap using Motive^®^ infrared cameras with a recording frequency of 120 Hz. The orders of magnitude of the head and head velocity were the main results extracted for several semi-aquatic and terrestrial snakes. 

Kinematic data were obtained from living animals swimming on a swimming raceway, thus with muscle actuation all along the body [29]. However, the 3D-printed snake body was inert and not fitted with actuators. Thus, we used dead snakes as biological inert models (non-intentionally killed and frozen until use; a permit was issued to X.B. to collect the dead snakes, DREAL 261679862017). The resulting ripple cones were compared under identical initial conditions (linear axis amplitudes and speeds). A snake robot of 1.2 m long with 45 vertebrae (disks) and 3D printed using Nylon (PA12) with MJF (Multi Jet Fusion) HP^®^ was compared with two dead snakes (*Hierophis viridiflavus*, *Zamenis longissimus*) (see Table 1). Dead snakes were unfrozen shortly prior to the experiments (we note that freezing may have partly damaged the muscular structures). 

### 3.2. Protocol

The snakes were studied as follows:(1)Extracting the undulation cone required monitoring of the entire body while moving. Thus, 11 reflective markers (see Figure 16) were taped to every fifth vertebra from the head to the tail of the inert robot. On the snakes, seven markers were taped and homogeneously distributed along the snake bodies (each ~40 vertebrae). Undulations were recorded using motion capture.(2)Each robot and snake (*Hierophis viridiflavus*, *Zamenis longissimus*) (see Table 1) head was actuated for nine seconds at a constant speed of 0.9 m/s and with a constant head amplitude of 0.17 m.(3)The applied head amplitude of the robotic snake varied while maintaining a constant head speed of 0.9 m/s. In the first experiment, an amplitude of 0.2 m was applied to the head; the second time, an amplitude of 0.17 m was applied.

The results are presented and discussed in the next subsection.

### 3.3. Results

The snakes’ undulations were described and compared (see Figure 18, Figure 19 and Figure 20). The ripple comparison of the resulting pattern was performed graphically by examining the ripple cone (equivalent to the swimming cone described in [29]). 

The ripple cone obtained from dead snakes showed a significant decrease in the amplitude from the head to the tail (see Figure 18a,b) due to the plastic deformation of the musculoskeletal system that absorbed energy. The plastic deformation was likely due to the viscoelastic properties of muscles and tendons and the long deep freeze. Ice crystals that formed during freezing partly destroyed the muscle cells and altered the elasticity. However, in the case of *Hierophis viridiflavus*, the amplitude at the mid-body was slightly higher (0.2–0.3 m), forming a belly and two modest nodes at 0.2 mand 0.4 m. This suggested a greater degree of elasticity in the middle of the body. The ripple cone corresponded to an inert body, as the muscles were not active, unlike the swimming cone observed in living snakes (Figure 21).

The bio-inspired snake robot ripple cone showed a closer pattern compared with the snake swimming cones (Figure 21). The head amplitude was higher than the mid-body and tail amplitudes, but two nodes and a belly at the mid-body were well-defined (see Figure 19b). This behavior likely resulted from torsion beam elasticity. Moreover, plastic deformation was not visible. The measured transversal speed drastically decreased from the head to the middle of the body but remained constant from the mid-body to the tail.

Reducing the head amplitude and increasing the head frequency while keeping the same input head speed resulted in a significant modification of the ripple cone (see Figure 20b) and the associated velocities distribution along the snake body (see Figure 20a). Two nodes and bellies were well-defined; the movement of the head, mid-body, and tail displayed peculiarities. Unlike the previous test, the variation in velocity was higher. The tail speed was higher than that of the mid-body. As expected, the speed of the head remained the highest because it was actuated by the linear axis of the device.

## 4. Discussion and Future Work

### 4.1. Discussion

Swimming and ripple cones, which were respectively obtained using living swimming snakes versus inert snakes and non-actuated robots, provided a straightforward means of comparing how the undulation waves propagated along the body axis.

The swimming cone produced by the living snakes (anguilliform swimmers) presented in [29] was composed of two nodes and a belly at the mid-body (Figure 21a,b). The amplitude increased from head to tail, and the inflexion point changed from head to tail along the body. This amplifying pattern resulted from the successive actuation of locomotor muscles along the body [33]. Indeed, the transverse velocity increased from the head to the tail with a maximal value at the mid-body (see Figure 22), suggesting a maximal muscular effort in the second portion of the body before the tail.

By actuating only the head of the inert snakes (dead snakes and robot), we also obtained propagating undulations. However, we observed decreasing amplitude patterns. Thus, the resulting ripple cone of dead snakes was substantially different from the swimming cone. This was likely explained by the absence of a muscular actuation propagating along the snake’s body (this was the main difference between a living and dead snake in this study framework). The amplitude variation, which is described by an amplitude ratio $\frac{h_{a}}{t_{a}}$, defined by the head amplitude $h_{a}$ and the tail amplitude $t_{a}$, was high for swimming snakes and low for dead snakes (see Figure 23). 

The ripple cone obtained with the snake robot was somehow intermediate to the swimming cones obtained with living snakes and the ripple cone obtained with dead snakes. According to the amplification of the bending moment, the snake robot’s body stiffness was higher than that of the inert snakes, which required a higher input head frequency than living snakes to obtain an undulating cone. The number of nodes and mid-body belly were identical to what was observed in swimming snakes, and they were highly apparent in the robot, but the decreasing pattern obtained with the robot contrasted with the increasing patterns of living snakes. The robot produced markedly greater amplitudes of undulations of the tail than at the mid-body (Figure 20). We suggest that robot elasticity played a significant role in the production of this intermediate pattern. The bodies of dead or anesthetized snakes are extremely flexible since the bodies are adapted to coil tightly and adopt extremely variable positions. Locomotor muscles likely exert finely tuned tensions on the skeleton, with different patterns during different movements (e.g., climbing, crawling, or swimming); however, this topic is out of the scope of the current study. Focusing on swimming in a straight line, the elasticity of the robot may have partly mimicked the tension of the whole body adopted by living snakes. The head amplitudes of the dead snakes and the robot were greater than those observed in biological snakes (see Figure 17a), especially compared with *Natrix maura*. Thus, the amplitude ratio was significantly higher in the inert robot compared with living snakes (see Figure 23). The head velocity we applied in the inert objects (dead snakes and robot) to generate the pattern with two nodes and a mid-body belly was notably higher than those measured on biological snakes with 0.9 m/s compared with 0.1 m/s in swimming snakes.

Overall, the comparisons between living, dead, and robot snakes suggested that the anguilliform swimming motion appeared to result from a combination of body stiffness and muscle actuation sequences. Stiffness appeared to be a significant factor in the stiffness/muscle actuation ratio. In fact, to minimize energy expenditure during swimming, it was necessary to reduce the number of actuators and to adjust their placements to optimize undulation and to mimic swimming cones with a relatively modest snake head velocity and amplitudes, followed by an amplification of the undulation amplitude (accompanied by greater transversal acceleration), at least until the mid-body. The snake robot skeleton elasticity that allowed the rough mimicking of swimming cones with a single actuator at the head provided an encouraging starting point. Indeed, actuation would only have to amplify the natural tendency of the robot to undulate like a snake. Optimizing the skeleton stiffness will permit a reduction in the number of actuators compared with biological snakes, with the final goal of achieving anguilliform swimming locomotion. 

### 4.2. Future Works

The snake robot was based on a precise analysis of a large amount of data collected on a diverse range of species of living and dead snakes. The artificial “skeleton” of the robot generated lateral undulations that resembled those observed in living snakes with minimal actuation (only the head). The Cosserat rod theory, coupled with the Lighthill model [22] for cable actuation, will be developed to improve the shape deformation of the snake robot as a function of varying parameters. The model will also be used as a basis for optimizing the stiffness of the snake robot to achieve the correct undulations according to the actuator’s positions.

Since the bio-inspired skeleton was made of different CU joints, an equivalent flexible beam should be fitted to model the deformations in a volume. In fact, observations showed that as biological snakes swam in a volume, the body undulated in two orthogonal planes (yet essentially in the horizontal plane). A dimensional optimization method should be implemented to size an equivalent non-linear elastic beam that deformed in a volume with a constant Young’s modulus. 

Our results clearly showed that the anguilliform undulations of living snakes could not be reduced to the actuation of the head (as expected). Instead, the observations suggested that most of the thrust was generated mid-body, precisely where the snake’s diameter is greater, and hence, where most of the muscular mass is present. Before placing actuators and cable attachments inside the snake robot, an experimental study will be performed to define the best positions of the highly simplified artificial musculature. An analytical–experimental–biological analysis and tests will be performed in this endeavor. This study will complete the BIM method. Furthermore, the body shape response according to frequency and amplitude, as well as speed, will be studied to address the dependency of these three parameters.

Finally, no drag nor buoyancy issues were addressed in this study, as the snake robot undulations were studied on a sloped table. A waterproof coating (i.e., artificial skin) will be needed to study snake body deformation on the water’s surface and underwater. The properties of the surface boundaries between the snake body and the water will be investigated. Thus, we plan to develop a skin. Then, we intend to test the snake robot in water.

## 5. Conclusions 

This paper presents a bio-inspired method (BIM) for synthesizing biology in robots. Three major steps were introduced. (1) The pseudo-rigid body method defined the general shape of a vertebra and the degrees of freedom. (2) The geometric shape was sized according to biological snakes and measurements. (3) The CU joint was sized according to the deformation of a dead snake body. The entire snake robot was modeled as a theoretical beam with a variable diameter and a constant Young’s modulus. Finally, a snake robot “skeleton” was printed, and its undulating behavior was compared with what was observed in dead snakes on a specifically developed testing bench. As a scientific tool, the snake robot was designed to study forward locomotion, but its maneuverability was not considered in this study. The key contributions of this paper can be summarized as follows:(1)A general comprehensive bio-inspired method for synthesizing specific locomotion was introduced. A BIM was applied to the development of a snake robot to perform undulations. Snake robot mechanical behavior was implemented directly from biological snakes, unlike traditional robots.(2)A new design to synthesize a snake vertebra based on a compliant universal joint was introduced. The CU joint reproduces the motions in a volume, ensuring stiffness according to the two respective plans. Each joint in a respective plan can be modeled as a beam with a constant Young’s modulus and a variable diameter. The global deformation of the snake robot body realizes fluid undulation, which differs from a traditional snake robot endowed with rigid modules.(3)The collaboration between biology and robotics led to a comparison of the snake’s behavior using an original testing bench. A comparison of the undulation cones demonstrated that internal actuation (muscles) combined with musculoskeletal system stiffness mostly occurred in the mid-body, where the amplitudes were the greatest. The direct biomimetic investigation between the flexible skeleton of the robot and biological snakes provides answers to biologists and a feedback loop to better understand how snakes move, for instance, which part of the snake should be targeted in future investigations. The snake robot fulfills this central role.

## Figures and Tables

**Figure 1 biomimetics-07-00223-f001:**
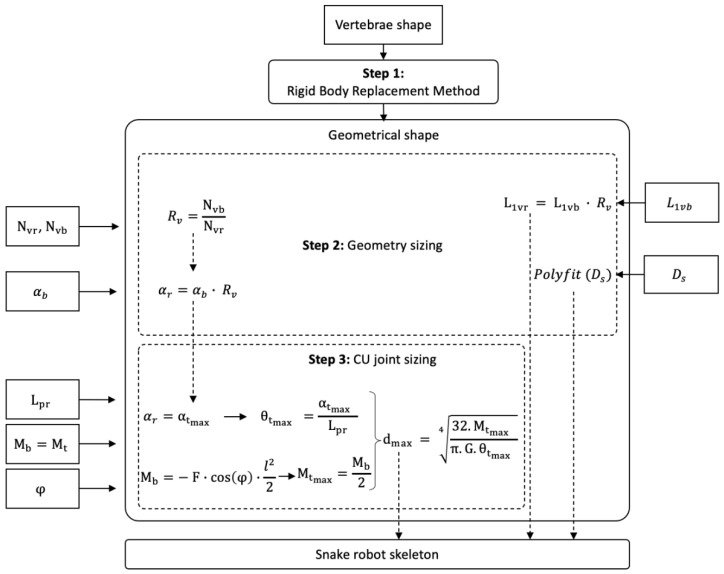
Bio-inspired method.

**Figure 2 biomimetics-07-00223-f002:**
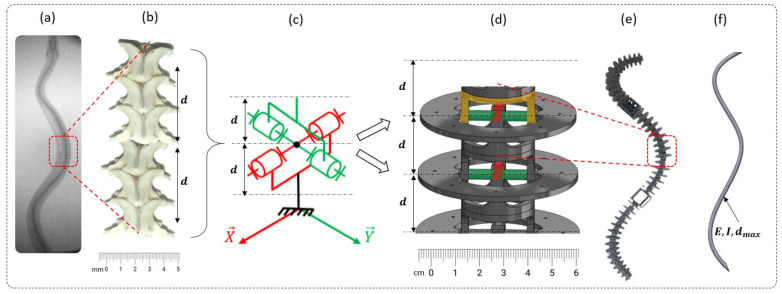
Rigid body replacement (RBR) method. (**a**) X−rays snake skeleton. (**b**) Skeleton portion (6 vertebrae). (**c**) Equivalency to one universal joint. (**d**) Flexible module. (**e**) Snake robot skeleton. (**f**) Equivalent theoretical beam. d is the distance between two vertebrae.

**Figure 3 biomimetics-07-00223-f003:**
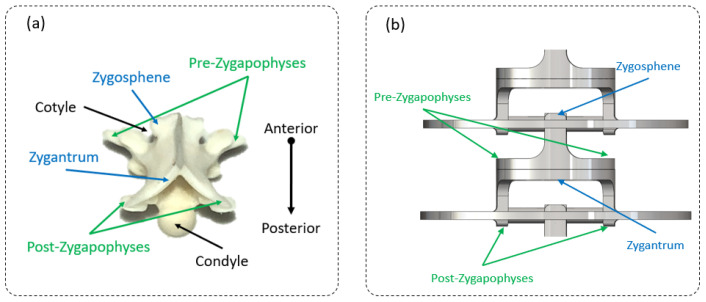
(**a**) Biological snake vertebrae (*Natrix natrix*). (**b**) Equivalent snake robot flexible vertebrae.

**Figure 4 biomimetics-07-00223-f004:**
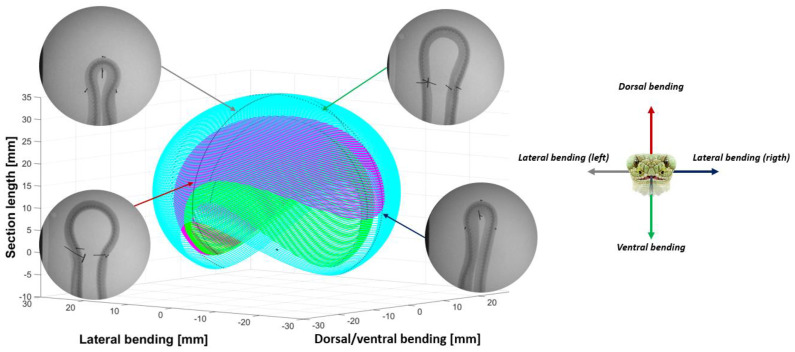
Neck swept area quantified using X-rays (magenta: *Python regius* 1, green: *Python regius* 2, cyan: *Helicops angulatus*).

**Figure 5 biomimetics-07-00223-f005:**
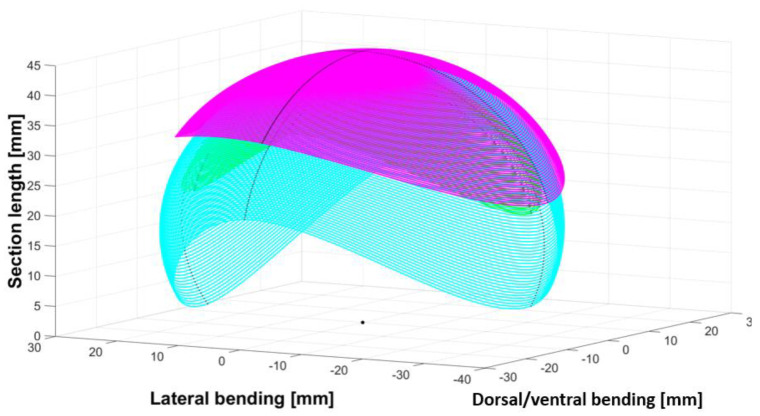
Middle body swept area quantified using X−rays (magenta: *Python regius* 1, green: *Python regius* 2, cyan: *Helicops angulatus*).

**Figure 6 biomimetics-07-00223-f006:**
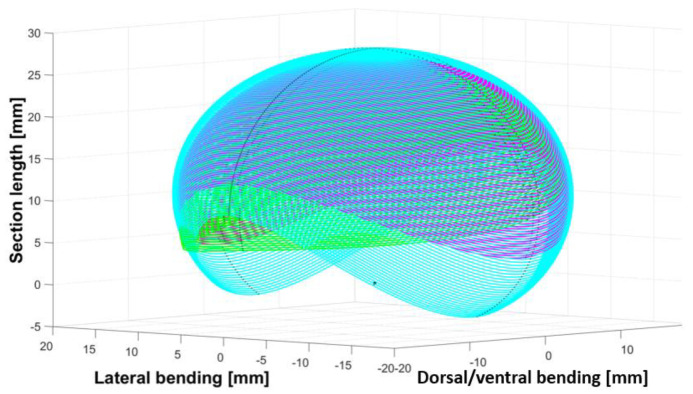
Tail swept area quantified using X−rays (magenta: *Python regius* 1, green: *Python regius* 2, cyan: *Helicops angulatus*).

**Figure 7 biomimetics-07-00223-f007:**
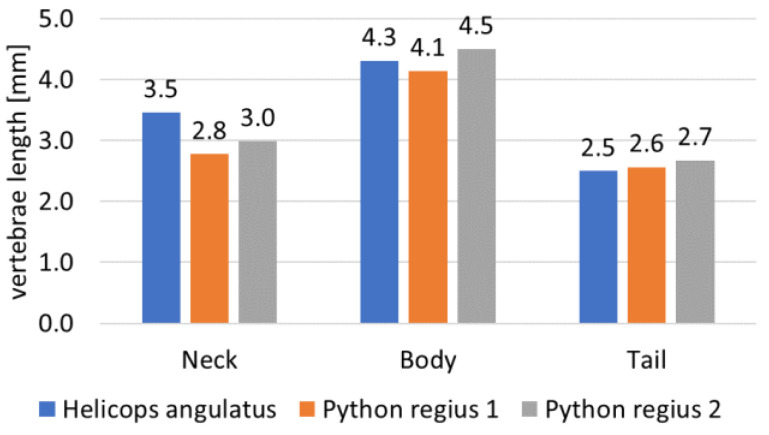
Vertebrae length measured using X-rays.

**Figure 8 biomimetics-07-00223-f008:**
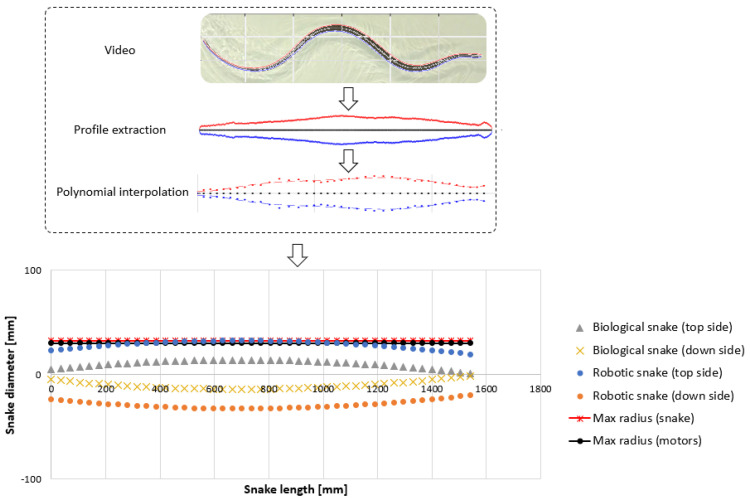
Snake robot diameter according to the profile extracted from the swimming snake (*Hierophis viridiflavus*). An offset of 15 cm was applied for the future placement of the actuators.

**Figure 9 biomimetics-07-00223-f009:**
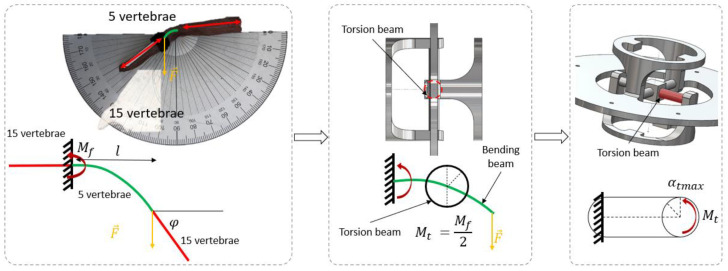
Method to pass from the equivalent bending moment to the equivalent torsion moment for lateral side deflection.

**Figure 10 biomimetics-07-00223-f010:**
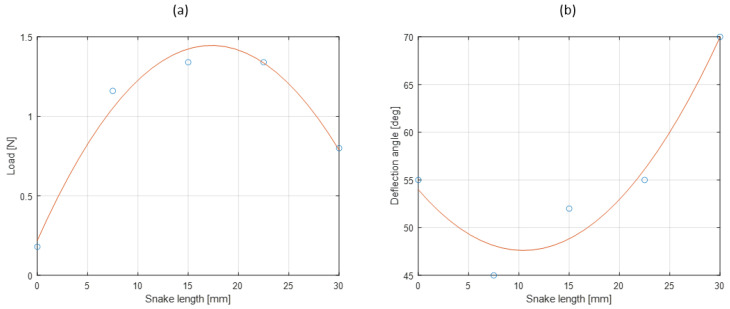
(**a**) Second-order polynomial force interpolation according to the snake length. (**b**) Second-order polynomial angle interpolation according to the snake length. Force and deflection angles were measured at five points on a *Hierophis viridiflavus* specimen.

**Figure 11 biomimetics-07-00223-f011:**
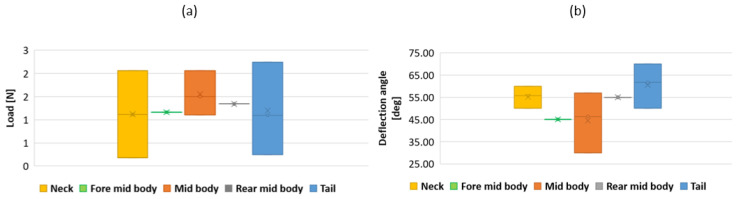
(**a**) Force measurement interval according to the snake position. (**b**) Deflection angle measurement interval according to the snake position. Force and deflection angles were measured using two *Hierophis viridiflavus* and one *Zamenis longissimus* specimen.

**Figure 12 biomimetics-07-00223-f012:**
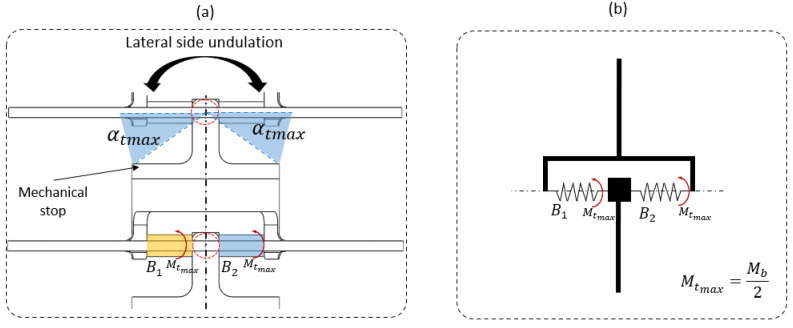
(**a**) Compliant joint with an elastic beam. Red circle represents the torsion beam in the normal plan. (**b**) Torsion beam equivalent to springs assembled in parallel. Arrows are the moment of torsion.

**Figure 13 biomimetics-07-00223-f013:**
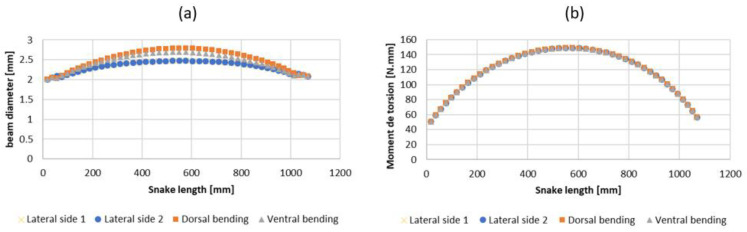
(**a**) Compliant beam diameters as a function of snake length. (**b**) Torsion applied to the beam as a function of the snake length.

**Figure 14 biomimetics-07-00223-f014:**
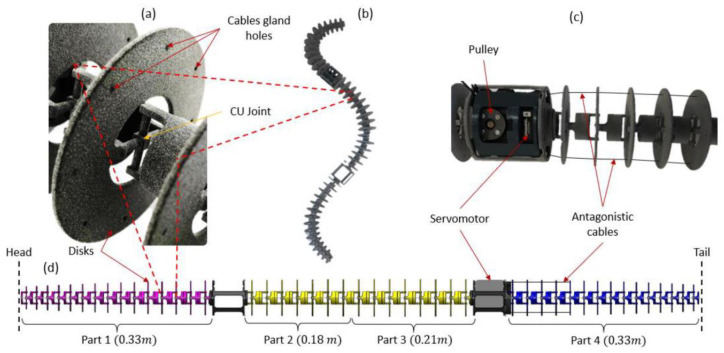
(**a**) Three-dimensional-printed compliant joint (PA12). (**b**) Three-dimensional-printed snake skeleton. (**c**) Description of cable routing for robot actuation. (**d**) Vertebral variation along the snake body.

**Figure 15 biomimetics-07-00223-f015:**
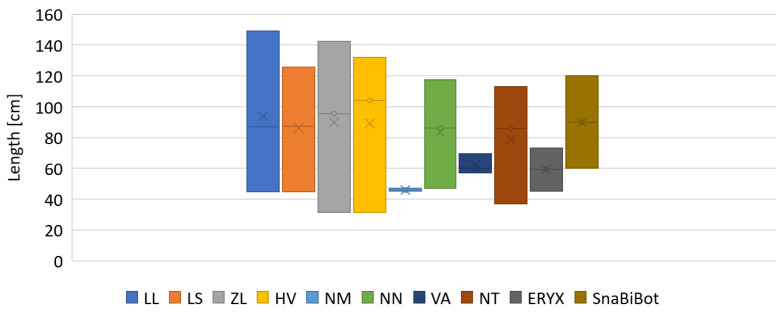
Snake length in centimeters. Sea snakes, LL: *Laticauda laticaudata*, LS: *Laticauda saintgironsi*; semi-arboreal snake, ZL: *Zamenis longissimus*; terrestrial racer, HV: *Hierophis viridiflavus*; semi-aquatic snakes, NM: *Natrix maura*, NN: *Natrix* natrix, NT: *Natrix tessellata*; terrestrial viper, *Vipera aspis*; fossorial sand boa, Eryx.

**Figure 16 biomimetics-07-00223-f016:**
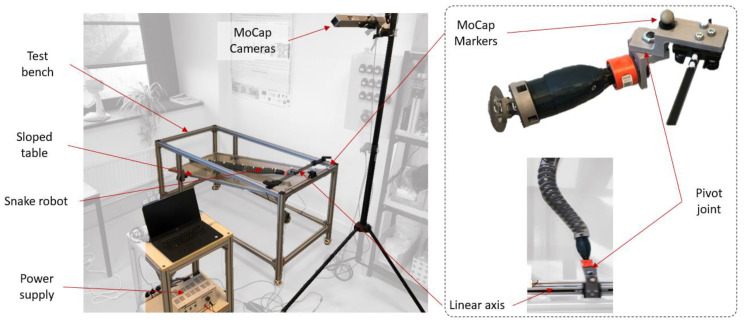
Test bench for snake robot and biological snake ripple.

**Figure 17 biomimetics-07-00223-f017:**
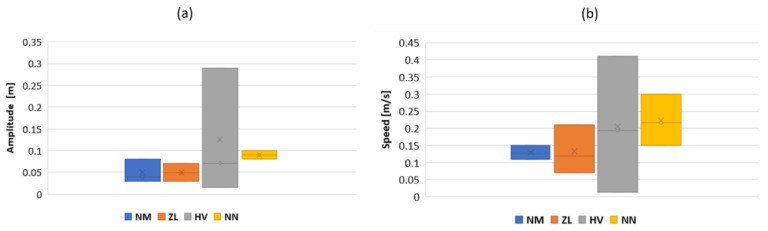
(**a**) Head amplitude range and (**b**) head speed range. NM: *Natrix maura*, ZL: *Zamenis longissimus*, HV: *Hierophis viridiflavus*, NN: *Natrix natrix*.

**Figure 18 biomimetics-07-00223-f018:**
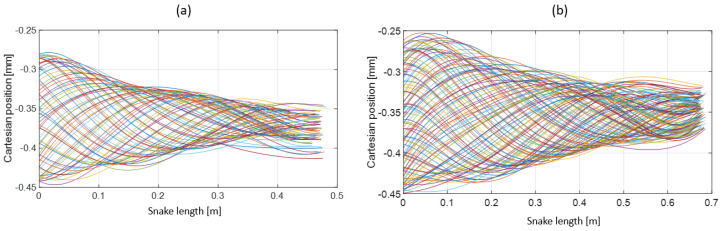
(**a**) Ripple cone from *Hierophis viridiflavus*. (**b**) Ripple cone from *Zamenis longissimus*. A length of 0 m corresponds to the head of the snake; the undulation amplitude decreased toward the tail. The colored lines show the body shape of the snake at each time for a whole undulatory sequence.

**Figure 19 biomimetics-07-00223-f019:**
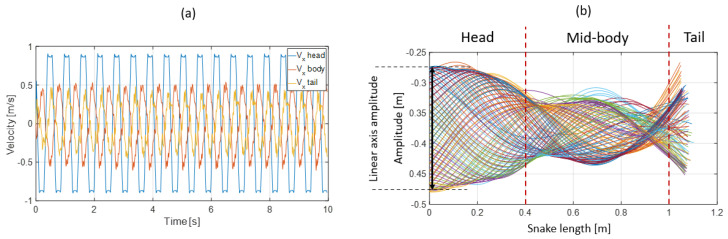
Snake robot skeleton response: table slope: 10.94 deg, head amplitude: 0.2 m, head speed: 0.86 m.s^−1^, middle body speed: 0.48 m.s^−1^, and tail speed: 0.4 m.s^−1^. (**a**) Snake robot skeleton head, middle body, and tail transverse speed. (**b**) Ripple cone (head actuation along a linear axis). See Appendix A.

**Figure 20 biomimetics-07-00223-f020:**
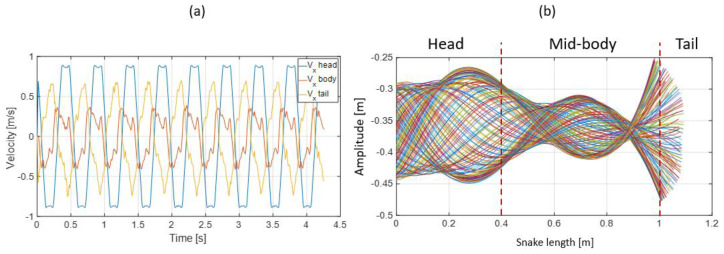
Snake robot skeleton response: table slope: 10.33 deg, head amplitude: 0.17 m, head speed: 0.86 m.s^−1^, middle body speed: 0.34 m.s^−1^, and tail speed: 0.69 m.s^−1^. (**a**) Snake robot skeleton head, middle body, and tail transversal speed. (**b**) Ripple cone (head actuation along a linear axis).

**Figure 21 biomimetics-07-00223-f021:**
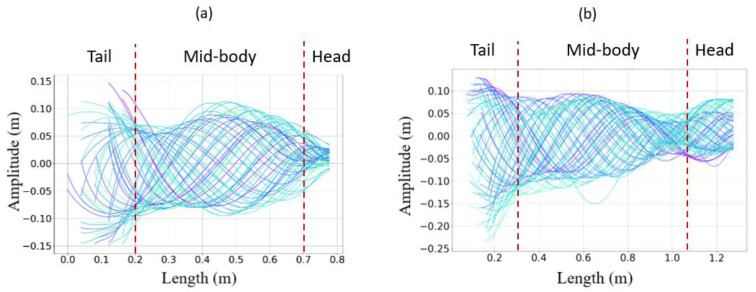
(**a**) *Natrix maura* swimming cone. (**b**) *Hierophis viridiflavus* swimming cone. The colored lines show the body shape of the snake at each time for a whole swimming sequence. The red dotted lines separate the tail from the body from the head of the snake.

**Figure 22 biomimetics-07-00223-f022:**
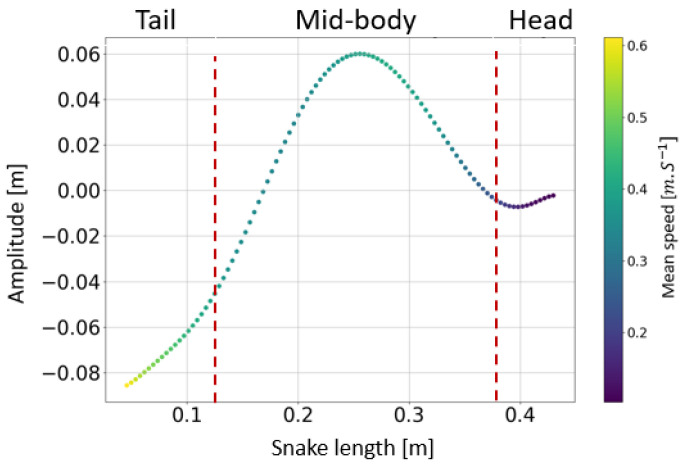
Variation in the transversal speed along the body in Natrix Maura. The tail was positioned on the left side and the head was on the right side. The red dotted lines separate the tail from the body from the head of the snake.

**Figure 23 biomimetics-07-00223-f023:**
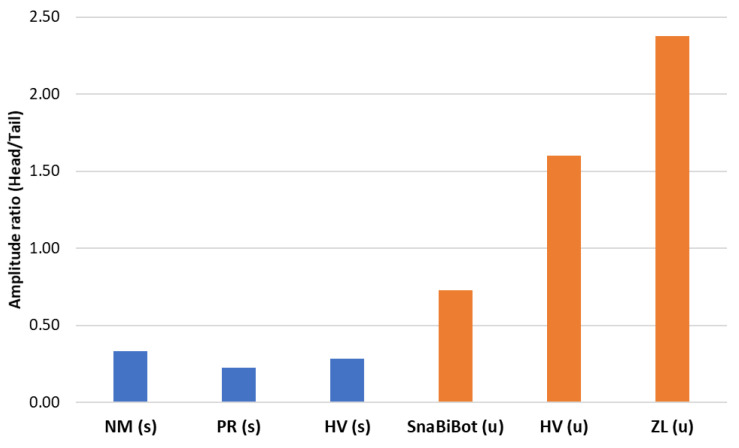
Snake ratio between the head amplitude and tail amplitude. SnaBiBot, HV, and ZL amplitude ratios were computed for a given speed of 0.86 m/s and a given amplitude of 0.17 m. (s) refers to swimming movement recorded in the raceway, while (u) refers to undulatory movements recorded in the test bench.

**Table 1 biomimetics-07-00223-t001:** Characteristics of the robotic and tested snakes.

Snake	Snout–Vent Length (m)	Body Mass (g)
SnaBiBot	1.2	220
*Hierophis viridiflavus*	0.9	202
*Zamenis longissimus*	0.8	196

## Data Availability

The datasets generated during and/or analyzed during the current study are available from the corresponding author upon reasonable request.

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
