# Peer review of "A Biomimetic Method to Replicate the Natural Fluid Movements of Swimming Snakes to Design Aquatic Robots"

_biomimetics, 2022, doi:10.3390/biomimetics7040223_

Round 1

Author Response

Please find attached the letter response to reviewers.

Reviewer 2 Report

The paper “A Biomimetic Method to Replicate Natural Anguilliform Movement to Design Bio-inspired Robots: Application to Swimming Snake Robots” by Gautreau et al. describes a novel bioinspired approach to implement locomotion information into bio-inspired robots. This method was used to develop a robot designed to reproduce lateral undulations of the whole body similar to a snake’s swimming patterns.

The article is quite interesting, but several crucial issues should be addressed before considering it suitable for acceptance.

The abstract should be reduced, and restructured providing a short statement of the context pf the study, the problem, what authors propose to solve the problem and method used, a summary of results obtained, and take-home message.

The introduction provides a long state of the art on biomimetics, often too long and not related to the study. For instance, including the birth of biomimetics, definitions, most popular cases of biomimetic products (e.g. Velcro, etc…). This Journal is on biomimetics, thus it is assumed that its readership knows very well fundamentals of biomimetics and bioinspiration.

However, many relevant works related to the prototype presented have not been included in the state pf the art. So, I strongly suggest authors to expand this part and reducing the redundant collection of biomimetic works not related to their robotic artifact.

For instance there are many examples of biomimetic undulatory swimming artifacts, reproducing fish, snake, or salamander patterns.

For example

Manfredi, L., Assaf, T., Mintchev, S., Marrazza, S., Capantini, L., Orofino, S., ... & Dario, P. (2013). A bioinspired autonomous swimming robot as a tool for studying goal-directed locomotion. Biological Cybernetics107(5), 513-527.

Ijspeert, A. J., Crespi, A., Ryczko, D., & Cabelguen, J. M. (2007). From swimming to walking with a salamander robot driven by a spinal cord model. Science315(5817), 1416-1420.

Also, there are recent works in which the undulation of a swimming biomimetic robot is achieved by cables, similarly to the robot presented in this study, but also including magnetic transmission system.

Romano, D., Wahi, A., Miraglia, M., & Stefanini, C. (2022). Development of a Novel Underactuated Robotic Fish with Magnetic Transmission System. Machines10(9), 755. https://doi.org/10.3390/machines10090755

Authors should include these works, comment on them, and comparing their robot with them. What novelty the proposed robot provides? This is a crucial point to address and to increase the scientific value of the study.

Also, in the design of the robot, the cost of transport of the robot has not been calculated. Also, there is no mention to the Strouhal number.

Drag and friction issues have been hastily mentioned, but not calculated.

Authors should include a section describing maneuverability of the robot. Can it just move forward or can also steer? What about buoyancy control?

A deep English revision is needed.

Author Response

(The authors gave the same response as above.)

Round 2

Reviewer 2 Report

Authors addressed much of my comments and the manuscript is much improved. I suggest authors to include an overview of the possible applications that these robotic fish or snakes have (scientifically and technologically).

For instance, they can be used as surrogate of the biological model to investigate biological features, or can be used to interact with biological organisms to study their behaviours, or can be used to develop artifact with novel locomotion ability compare to traditional robots.

Some useful examples

Crespi, A., Badertscher, A., Guignard, A., & Ijspeert, A. J. (2004). An amphibious robot capable of snake and lamprey-like locomotion. In Proceedings of the 35th international symposium on robotics (ISR 2004) (No. CONF).

Romano, D., Benelli, G., Hwang, J. S., & Stefanini, C. (2019). Fighting fish love robots: mate discrimination in males of a highly territorial fish by using female-mimicking robotic cues. Hydrobiologia833(1), 185-196.

Park, Y. J., Huh, T. M., Park, D., & Cho, K. J. (2014). Design of a variable-stiffness flapping mechanism for maximizing the thrust of a bio-inspired underwater robot. Bioinspiration & biomimetics9(3), 036002.

This part would increase the scientific value of the mansucript, but it has been overlloked by athors.

English editing still require revision.